# Condensin DC loads and spreads from recruitment sites to create loop-anchored TADs in *C. elegans*

Jun Kim[1†], David S Jimenez[1†], Bhavana Ragipani[1], Bo Zhang[2], Lena A Street[1], Maxwell Kramer[1], Sarah E Albritton[1], Lara H Winterkorn[1], Ana K Morao[1], Sevinc Ercan[1]*

[1]Department of Biology and Center for Genomics and Systems Biology, New York University, New York, United States; [2]UCSF HSW, San Francisco, United States

**Abstract** Condensins are molecular motors that compact DNA via linear translocation. In *Caenorhabditis elegans*, the X-chromosome harbors a specialized condensin that participates in dosage compensation (DC). Condensin DC is recruited to and spreads from a small number of *recruitment elements* on the *X*-chromosome (*rex*) and is required for the formation of topologically associating domains (TADs). We take advantage of autosomes that are largely devoid of condensin DC and TADs to address how *rex* sites and condensin DC give rise to the formation of TADs. When an autosome and X-chromosome are physically fused, despite the spreading of condensin DC into the autosome, no TAD was created. Insertion of a strong *rex* on the X-chromosome results in the TAD boundary formation regardless of sequence orientation. When the same *rex* is inserted on an autosome, despite condensin DC recruitment, there was no spreading or features of a TAD. On the other hand, when a '*super rex*' composed of six *rex* sites or three separate *rex* sites are inserted on an autosome, recruitment and spreading of condensin DC led to the formation of TADs. Therefore, recruitment to and spreading from *rex* sites are necessary and sufficient for recapitulating loop-anchored TADs observed on the X-chromosome. Together our data suggest a model in which *rex* sites are both loading sites and bidirectional barriers for condensin DC, a one-sided loop-extruder with movable inactive anchor.

*For correspondence:
se71@nyu.edu

†These authors contributed equally to this work

**Competing interest:** The authors declare that no competing interests exist.

## Editor's evaluation

This paper is likely to be of broad interest to researchers in the chromosome biology field. With specific loading sequences identified, the Condensin dosage compensation complex studied here provides an elegant system to investigate the in vivo activities of SMC complexes. Combining Hi-C, ChIP-seq and RNA-seq, the authors reveal that the complex spreads along the chromosome to create chromosome loops.

## Introduction

Eukaryotic chromosome structure is dynamically regulated across the cell cycle. During interphase, genomes are organized within the nucleus and go through further compaction for accurate segregation of chromosomes during mitosis and meiosis (*Gibcus et al., 2018*; *Himmelbach et al., 2018*; *Kumar et al., 2020*; *Rowley and Corces, 2018*; *Stam et al., 2019*; *Szalaj and Plewczynski, 2018*). Chromosome compaction is mediated in part by DNA looping by a conserved family of protein complexes called the structural maintenance of chromosomes (SMCs) complexes (*van Ruiten and Rowland, 2018*; *Kinoshita et al., 2022*; *Shaltiel et al., 2022*; *Wood et al., 2010*).

In vitro analysis of SMC complexes, including condensin and cohesin, indicates that they act as ATP dependent molecular motors forming DNA loops by progressively extruding DNA (*Çamdere et al., 2018*; *Ganji et al., 2018*; *Kong et al., 2020*; *Terakawa et al., 2017*). In silico modeling of loop extrusion activity to explain Hi-C data upon condensin- and cohesin-related perturbations provides strong support for loop extrusion hypothesis in vivo (*Gibcus et al., 2018*; *Nuebler et al., 2018*; *Fudenberg, 2017*).

In mammalian cells, interphase organization of chromosomes into topologically associating domains (TADs) is mediated by cohesin and its regulators. Cohesin loading and processivity are promoted by the Adherin complex (in yeast Scc2-Scc4, in mammals Nipbl & Mau2), and its unloading is mediated by Wapl (*Peters and Nishiyama, 2012*; *Kim et al., 2019*; *Haarhuis and Rowland, 2017*; *Gassler et al., 2017*). Cohesin translocation on DNA is also controlled by insulator proteins, including the zinc finger transcription factor CTCF, which creates TAD boundaries (*Nora et al., 2017*).

While in vitro and in silico experiments suggest similar molecular activities for condensins and cohesin, it is less clear how condensin binding and movement on eukaryotic chromosomes are regulated in vivo (*van Ruiten and Rowland, 2018*). In yeast, condensin regulates chromosomal interactions across the ribosomal DNA and form gene loops involved in repression of the quiescent genome (*Swygert et al., 2021*; *Paul et al., 2018*; *Kim et al., 2016*). However, since yeast chromosomes support much smaller interaction domains and lack clear TAD boundaries, how condensin binding regulates larger eukaryotic genomes remains unknown. An excellent model to address this gap is a specialized condensin that functions within the X-chromosome dosage compensation complex (DCC) in *Caenorhabditis elegans*.

In addition to the canonical condensins I and II, *C. elegans* possess condensin I$^{DC}$ (hereafter condensin dosage compensation [DC]) that differs from condensins I by a single subunit, the SMC-4 variant DPY-27 (*Csankovszki et al., 2009*). Condensin DC binds to both X-chromosomes in hermaphrodites to repress their transcription by a factor of two, equalizing overall X chromosomal transcripts between XX hermaphrodites and XO males (*Kruesi et al., 2013*; *Kramer et al., 2016*; *Kramer et al., 2015*; *Albritton and Ercan, 2018*). Several features make condensin DC a powerful system to address mechanisms of condensin binding and spreading. First, unlike canonical condensins, the sequence elements important for condensin DC recruitment to the X-chromosomes are identified (*McDonel et al., 2006*; *Jans et al., 2009*; *Ercan et al., 2007*; *Albritton et al., 2017*). Second, the spreading of the complex can be distinguished from recruitment using X to autosome (X;A) fusion chromosomes (*Ercan et al., 2009*). Third, since the complex only binds to the X-chromosomes, autosomes serve as internal controls, allowing sensitive measurement using genomics approaches (*Albritton et al., 2017*; *Vielle et al., 2012*; *Street et al., 2019*; *Ercan and Lieb, 2009*).

Condensin DC recruitment to the X-chromosomes is mediated by ~60 recruitment elements on the *X*-chromosome (*rex* sites) (*Jans et al., 2009*; *Ercan et al., 2007*; *Albritton et al., 2017*). A number of these elements was validated to autonomously recruit the DCC on extrachromosomal arrays (*McDonel et al., 2006*; *Jans et al., 2009*; *Csankovszki et al., 2004*). Furthermore, due to the repetitive nature of the extrachromosomal arrays, strong *rex* sequences depleted condensin DC from the X-chromosomes, demonstrating their recruitment activity (*McDonel et al., 2006*). Hi-C analysis in *C. elegans* embryos indicated that eight strong *rex* sites function as TAD boundaries and form long-range *rex-rex* loops on the X-chromosomes (*Rowley et al., 2020*; *Crane et al., 2015*). Recruitment of condensin DC leads to spreading to the entire chromosome, accumulating at hundreds of enhancers, promoters, and other accessible gene regulatory elements (*Street et al., 2019*).

Here, we addressed the mechanism by which *rex* sites and condensin DC form TADs. Without *rex* sites, spreading of condensin DC to the autosomal region of the X;V fusion chromosome increases 3D DNA contacts but fails to form TADs. Insertion of a *rex* site on X-chromosome sufficiently leads to the formation of a loop-anchored TAD irrespective of orientation, suggesting that *rex* sites are bidirectional barriers to loop extrusion. However, insertion of the same single strong *rex* to chromosome-II, which is largely devoid of condensin DC, did not form a TAD boundary, suggesting that *rex* sites are barriers largely specific to condensin DC. The insertion of '*super rex*' composed of an array of strong *rex* elements was capable of creating a spreading domain and forming a boundary between two domains of interaction on either side of the insertion. Insertion of three *rex* sites on an autosome led to recruitment and spreading of condensin DC and formed a loop-anchored TAD. Together our data

suggest that *rex* sites are both loading sites and bidirectional barriers for condensin DC, a one-sided loop extruder with movable inactive anchor.

## Results

## Condensin DC is a developmentally maintained loop-extruding factor targeted to the X-chromosome

We performed Hi-C analysis in hermaphrodite mixed developmental stage embryos and L3 larvae and describe three notable observations indicating that X-chromosome harbors an additional loop-extruding factor (LEF), condensin DC (*Figure 1A*), throughout the development of *C. elegans*.

First is the presence of TADs on the X-chromosome. As randomly distributed LEFs track along DNA, pairs of increasingly distant genomic loci are looped in 3D space except when LEFs encounter a barrier. This results in the observed insulation effect, whereby contacts within TADs are enriched more than contacts across TADs. In the mammalian system, the essential components of TAD formation consist of cohesin and CTCF, which correspond to LEFs and barriers, respectively (*Fudenberg, 2017*; *Nora et al., 2017*; *Schwarzer et al., 2017*). In *C. elegans*, while cohesin is present, a CTCF homolog is absent (*Heger et al., 2009*). Therefore, no strong TAD structures are present on autosomes (*Figure 1B*, chromosome-I) as previously reported (*Crane et al., 2015*). On the other hand, X-chromosome is globally bound by condensin DC and organized into distinct TADs separated by *rex* sites, which are strongly bound by the DCC subunits, including SDC-2 and SDC-3 (*Figure 1B*). Unlike the CTCF removal experiments where cohesin remains bound, the loss of SDC-2 results in the failure of condensin DC to localize to the X-chromosome and dissolution of TADs, suggesting that SDC-2 function as both LEF loader and barrier (*Crane et al., 2015*; *Anderson et al., 2019*; *Pferdehirt et al., 2011*).

Second, X-chromosome show a characteristic enrichment of 3D DNA contacts compared to autosomes. The P(s) and its log-derivative are powerful metrics of Hi-C data allowing interpretation of DNA loops (*Fudenberg, 2017*; *Gassler et al., 2017*; *Polovnikov et al., 2022*). Experimental perturbations of cohesin and its regulator proteins (NIPBL/WAPL) in the mammalian system led to the conclusion that the decrease in the steepness of the P(s), or the characteristic shoulder, observed in the unperturbed state is indicative of loop extrusion (*Fudenberg, 2017*; *Haarhuis and Rowland, 2017*; *Nora et al., 2017*; *Schwarzer et al., 2017*). Here, we observe the similar characteristic shoulder for the X-chromosome compared to the autosomes (*Figure 1D*). LEF modeling of Hi-C data proposes that the peak position on the log-derivative of P(s) corresponds to the mean loop size and that increasing the number of LEFs in loop dense regime would decrease the mean loop size (*Gassler et al., 2017*; *Polovnikov et al., 2022*; *Goloborodko et al., 2016*). Using the same metric, we find that the mean loop size for *C. elegans* X-chromosomes is 200 kb in embryo and 125 kb in L3s and 400 kb for autosomes. The X-chromosomes thus have a smaller loop size than autosomes (*Figure 1D*). Consistent with condensin DC being an LEF, the loss of SDC-2 abolishes both condensin DC binding and this characteristic shoulder (*Crane et al., 2015*; *Anderson et al., 2019*).

Third, the X-chromosome shows weaker A/B sub-compartmentalization than autosomes. The checker-board pattern of the Hi-C matrix, also known as A/B compartmentalization, indicates the spatial segregation between transcriptionally active and inactive regions of the genome (*Imakaev et al., 2012*). The loss of cohesin or the cohesin-unloader WAPL results in respective increase or decrease in compartmentalization, suggesting that the activity of cohesin regulates compartmentalization (*Nuebler et al., 2018*; *Haarhuis and Rowland, 2017*; *Nora et al., 2017*; *Schwarzer et al., 2017*). Previous compartment analysis in *C. elegans* has shown that each chromosome is largely divided into three sections: two flanking arms and the center (*Crane et al., 2015*; *Bian et al., 2020*). This division has been assigned the term A/B compartments, with the center being the singular A segment and the two arms making up the two segments of B compartment. However, upon zooming in, we observe a more fine-scale checker-board pattern that coincides with orthogonal data such as H3K27ac/me3 and chromHMM (*Figure 1B and E*). The comparison of autosomes and the X-chromosome reveals that the sub-compartment strength of X-chromosome is weaker than that of autosomes (*Figure 1E*, *Figure 1F*, *Figure 1—figure supplement 1*). This is consistent with X-chromosome harboring additional LEFs, as additional active movement of chromatin imposed by condensin DC would further antagonize compartmentalization.

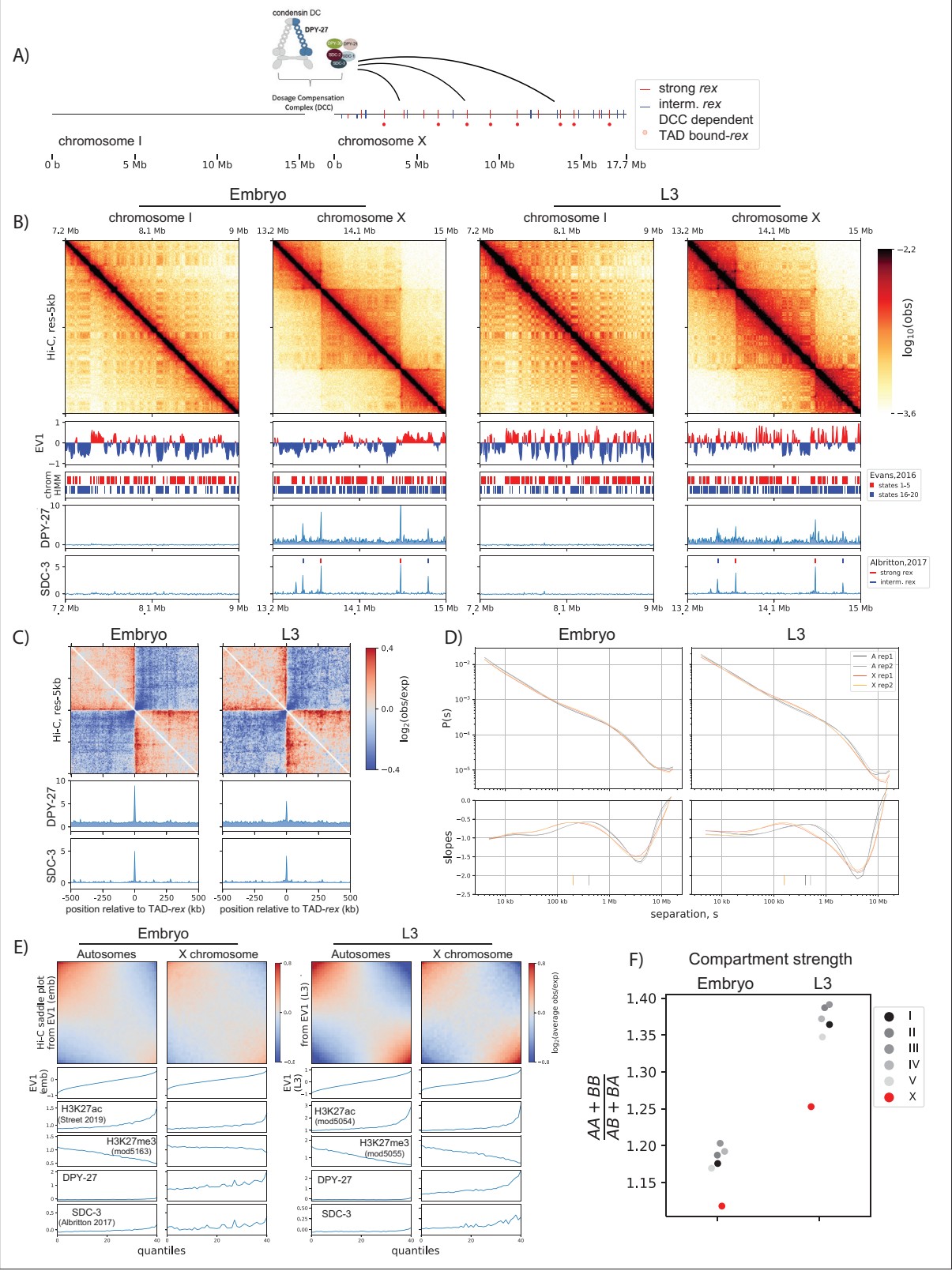

**Figure 1.** Developmentally conserved evidence of X-chromosome enriched loop extruding factor, condensin dosage compensation (DC). (**A**) Cartoon of the dosage compensation complex in *C. elegans*. Strong and intermediate recruitment elements on the *X*-chromosome (*rex*) sites (***Albritton et al., 2017***) and DC complex (DCC)-dependent topologically associating domain (TAD) boundary *rex* sites (***Anderson et al., 2019***) are annotated. (**B**) Snapshot of chromosome-I and X for embryo and L3. EV1 is computed using the center regions of chromosomes as defined in ***Ikegami et al.,***

*Figure 1 continued*

*2010*. The center regions have lower LEM-2 association, higher gene-density, and lower repeat density than the arms. Shown below are annotated ChromHMM states 1–5 and 16–20 that are associated with highest and lowest quantiles of gene expression, respectively (*Evans et al., 2016*). SDC-3 and DPY-27 binding is plotted as normalized Chromatin immunoprecipitation followed by high-throughput sequencing (ChIP-seq) scores. (**C**) Meta-plot of observed Hi-C interactions compared to expected, centered across strong *rex* sites. The expected values are computed using only the X-chromosome due to X and autosomes having distinct P(s) curves. (**D**) Hi-C relative contact probability, P(s), as a function of genomic separation, s, and the log-derivative of P(s) for two biological reps. The local-maxima of slopes or inferred mean loop size is annotated as vertical ticks. (**E**) Compartmentalization of all autosomes and the X-chromosomes at the center region of the chromosomes. The saddle plot indicates the level of intrachromosomal interactions between and within A and B compartments defined as top and bottom halves of EV1 values. (**F**) The strength of compartmentalization was calculated for each chromosome by taking a ratio of the sum of interaction within the same compartment to the sum of interactions across compartment as described in *Nuebler et al., 2018*. Compartmentalization is weaker in mixed developmental stage embryos, compared to L3 larvae. In both developmental stages, compartmentalization is weaker on the X. See *Figure 1—figure supplement 1* for an alternative method of computing compartment strength.

The online version of this article includes the following figure supplement(s) for figure 1:

**Figure supplement 1.** The strength of compartmentalization is computed using a sliding window submatrix (related to *Figure 1E, F*).

In summary, the strong *rex* sites acting as TAD boundaries, characteristic shoulder in contact frequency plots, and weaker sub-compartments of the X-chromosome suggest that condensin DC is an LEF that binds to and organizes X-chromosome structure.

## Spreading of condensin DC entails loop extrusion but cannot sufficiently form TADs without *rex* sites

In the mammalian system, the two components of TAD formation are genetically separable, namely cohesin and CTCF. This allows one to study the effect of LEFs in the absence of barriers by removing CTCF. However, in the *C. elegans* DC system, a protein that uniquely corresponds to the barrier component of the model has not been isolated. Therefore, to address the function of condensin DC in the absence of barrier elements, we performed Hi-C on a strain in which chromosome-X and V are fused, where condensin DC 'spreads' into chromosome-V, which does not have any *rex* sites (*Ercan et al., 2009*).

We make three observations on the Hi-C matrix. First, there are no noticeable TADs on chromosome-V side of the fusion (*Figure 2A*), suggesting that *rex* sites are unique elements of the X-chromosome that are required for TAD formation. Second, enrichment of DNA contacts closer to the main diagonal occur specifically for the chromosome-V side of the fusion (*Figure 2A*, third panel), which coincides with the decrease in mean loop size (*Figure 2B*). Third, weakening of checker-board pattern on chromosome-V side (*Figure 2A*, inset), which coincides with the local decrease in sub-compartment strength (*Figure 2C*). Hi-C features on the chromosome-V side becoming more similar to the X-chromosome are consistent with the generalized model that the activity of LEFs, and not the presence of barriers, modulates chromosome loop size and compartments. In summary, the spreading of condensin DC is partially a reflection of loop extrusion, which cannot sufficiently create TADs without the presence of *rex* sites.

## *Rex* sites are bidirectional barriers for condensin DC loop extrusion

The stalling of cohesin translocation by CTCF is directional, in which two convergent CTCF binding motifs brought together by loop-extruding cohesin (*Nanni et al., 2020*; *Nichols and Corces, 2015*; *Nishana et al., 2020*; *Rao et al., 2014*). We hypothesized that the DCC system may share mechanistic similarities through a CTCF-like protein. While previous work noted the lack of orientation bias in *rex-rex* corner-peaks (observed as pronounced 'dots' at the corners of TADs) (*Rowley et al., 2020*; *Anderson et al., 2019*), such computational inference has limitations due to the small number of looping *rex* sites and most *rex* sites having multiple motifs oriented in both directions. Thus, we experimentally tested the directionality of the 12 bp *rex* motifs in its ability to block loop extrusion by condensin DC. We chose *rex-8*, which is a TAD boundary and contains four copies of the 12 bp motif all oriented in the same direction. We inserted *rex-8* in the same position on the X-chromosome in opposite orientations. The *rex-rex* corner-peaks and the new boundary on either side of the inserted *rex-8* were similar regardless of the orientation of insertion (*Figure 3*). Therefore, unlike CTCF binding sites, *rex* sites are bidirectional barriers to loop extrusion.

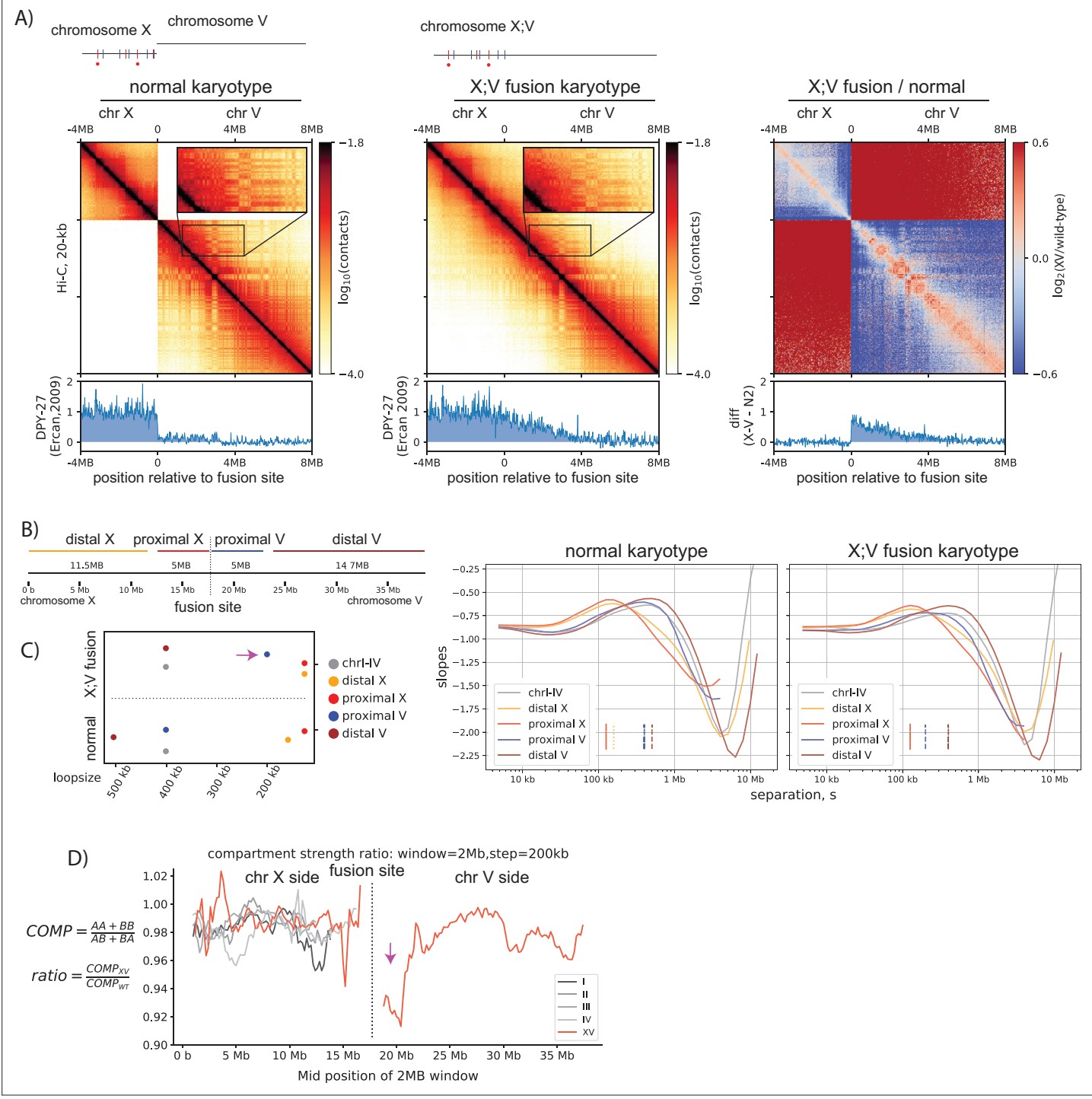

**Figure 2.** Spreading of condensin dosage compensation (DC) coincides with decrease in loop size and weakening of compartments. (**A**) Condensin DC spreads ~3 MB into the autosomal region of X;V fusion chromosome. 12-MB region snapshot of Hi-C contacts in the wildtype karyotype and X;V fusion strain. Hi-C and ChIP-seq data are from embryos. The inset zoom (10 kb-binned matrix) highlights weakening of the checker-board pattern indicating compartmentalization. Log2ratio plot highlights enrichment (red) of 'shorter-range' interactions at the spreading region of X;V. (**B**) To compare the local effect of spreading, chromosome X;V is divided into four regions surrounding the fusion site and log-derivative of P(s) is plotted. (**C**) Inferred mean loop size of each region from the derivative plot is plotted. Purple arrow highlights the decrease in mean loop size for chromosome-V side of the fusion (proximal V). (**D**) Difference in local compartment strength in X;V compared to normal karyotype. EV1 and saddle strength are computed for each 2-MB submatrix. The ratio of the same region between two conditions highlights distance-dependent decrease (purple arrow) in compartment strength on chromosome-V side of the fusion.

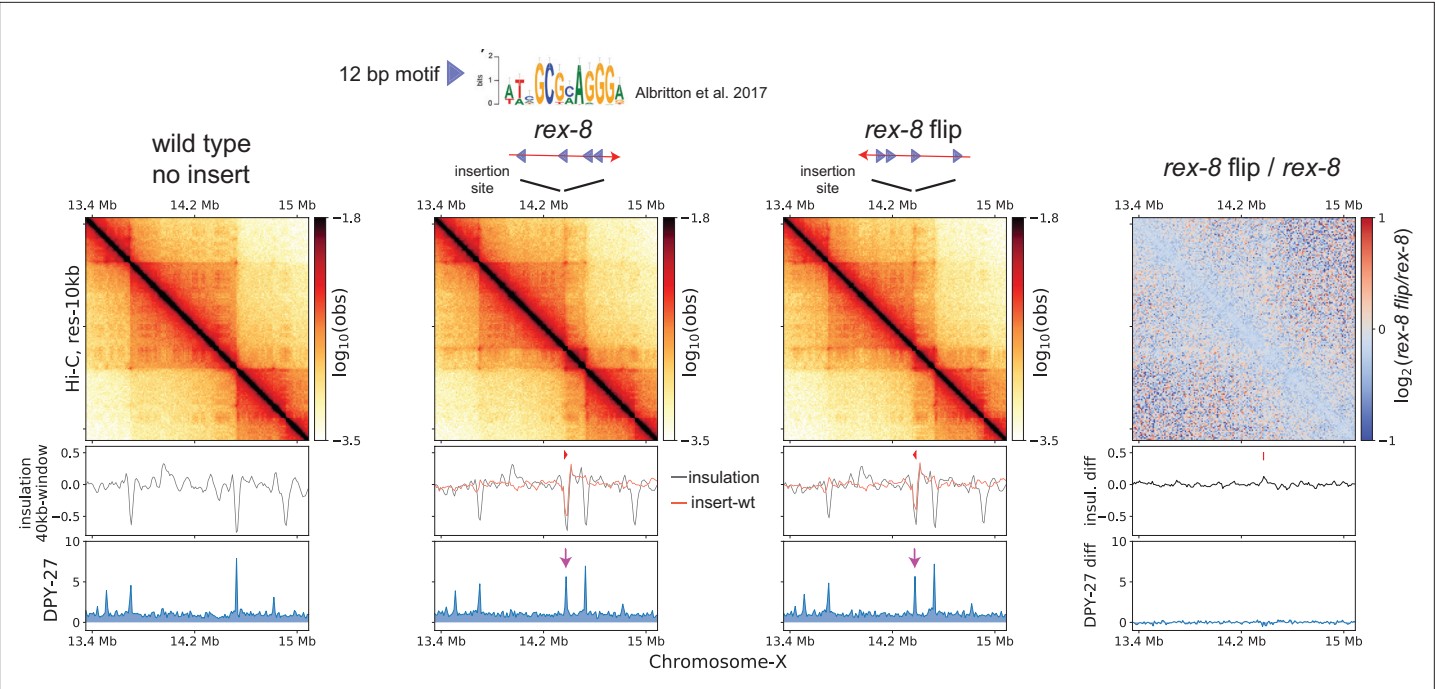

**Figure 3.** Ectopically inserted *rex-8* in two different orientations act as a bidirectional barrier. A strong recruitment elements on the *X*-chromosome (rex) site with four motifs in the same orientation acts as a topologically associating domain (TAD) boundary on the X. The 358-bp *rex-8* was inserted in two orientations at the same site on X-chromosome. Hi-C interactions surrounding the insertion sites along with the insulation score (black) and its subtraction to that of wild type (red), and DPY-27 ChIP-seq are plotted. A dip in the insulation score coincides with enrichment of DPY-27 ChIP (purple arrow) and supports barrier activity. Log2ratio plot highlights the lack of noticeable Hi-C difference between insertion of two different orientations.

## A dCas9-based block failed to recapitulate *rex*-like boundary on the X-chromosome

The *rex* insertion on X-chromosome suggests that the *rex* sites are physical blocks at which condensin DC stalls. A previous in vitro study showed that linear translocation of cohesin can be blocked using a dCas9-mediated protein obstacle (**Stigler et al., 2016**). To test whether a large block on the chromatin fiber could prevent condensin DC translocation along chromatin, we utilized a dCas9 based system targeting a repetitive region of the X-chromosome (**Figure 4**). The relative size of each individual protein complex (dCas9-SunTag+Pfib-1:NLS:scFv-GFP) is approximately 1400 kDa and is predicted to be larger than the ~20 nm blocks used for in vitro experiments (**Stigler et al., 2016**). In addition to the size of the block being large enough to block spreading, we utilized a repetitive region as the target so that multiple blocking complexes could be recruited to a relatively small genomic locus.

To characterize the effect of the block on condensin DC binding, we performed DPY-27 ChIP-seq and normalized coverage to input (no immunoprecipitation) and IgG (non-specific antibody) controls (**Figure 4**). While the input normalized ChIP-seq data showed condensin DC binding specifically in the presence of the sgRNA, a similar enrichment was observed upon IgG ChIP (**Figure 4—figure supplement 1**). This is not an artifact of sequencing bias in the input since the input tracks remain unaffected in the presence of sgRNA (**Figure 4—figure supplement 1**). Furthermore, this increased 'chippability' at the target locus is specific to antibodies and not due to cross-reactivity with protein A or G. Therefore, we refrain from interpreting ChIP-seq data, as targeting of dCas9-SunTag renders the locus susceptible to being precipitated by any antibody.

We instead turned to Hi-C data to bypass the use of antibodies and infer the barrier effect of dCas9 block. This too was complicated by the fact that X-chromosome is defective in SunTag strain (**Figure 4—figure supplement 2**), which results in a higher proportion of males in the population. This in turn leads to weaker TADs on the X-chromosomes due to males lacking DCC. Nevertheless, relative to the surroundings, the targeted locus shows enhanced insulation effect, suggesting that the dCas9 block insulated interactions across the target site.

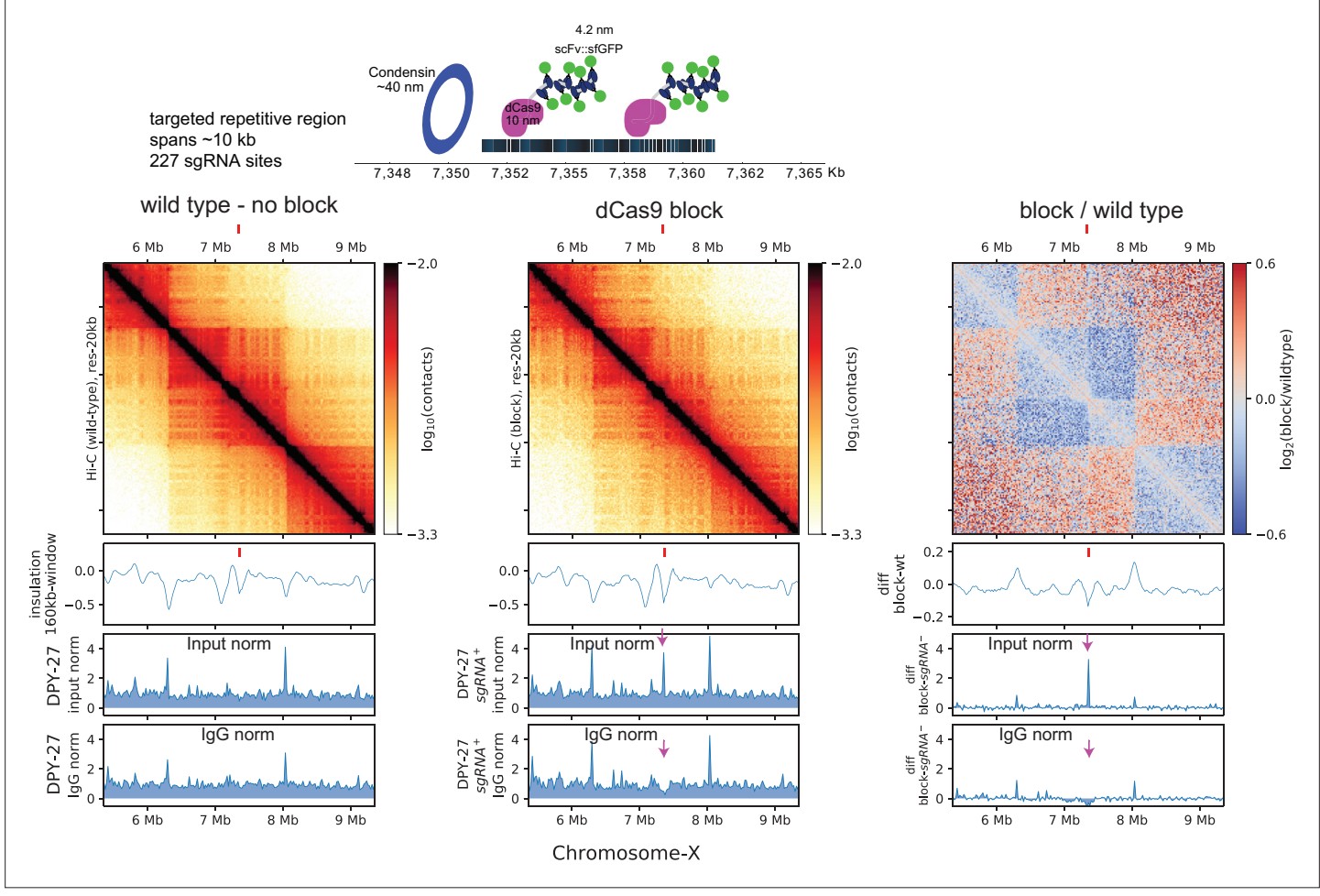

**Figure 4.** dCas9 block fail to sufficiently recapitulate *rex*-like boundary on the X-chromosome. Top: schematic depicting the multi-protein block and the approximate size of the components utilized to prevent condensin from translocating linearly along chromatin. Bottom: Hi-C of wildtype and block strain expressing all components of dCas9-SunTag and single guide RNA (sgRNA). The tick marks point to the block target. The below are ChIP-seq data in dCas9-SunTag expressing strain with (first panel) and without (middle panel) sgRNA. The two ChIP-seq tracks show normalization by input and IgG. Arrows pointing down to the DPY-27 ChIp signal that is apparent when data is normalized to input but not to IgG.

The online version of this article includes the following figure supplement(s) for figure 4:

**Figure supplement 1.** Increased 'chippability' due to dCas9-SunTag targeting.

**Figure supplement 2.** Global effect of Hi-C data in dCas9-SunTag targeting strain.

## Condensin DC is loaded at *rex* sites and spreads in either direction

A previous model for condensin DC-mediated organization of X-chromosome structure was derived from cohesin/CTCF: condensin DC initiates loop extrusion until it encounters a *rex* site to form a TAD boundary (*Fudenberg, 2017*; *Anderson et al., 2019*). One key assumption made in this model is that condensin DC loads uniformly throughout the X-chromosome. This assumption is contradicted by previous work demonstrating that large chunks of non-*rex* DNA from the X-chromosome fail to recruit condensin DC on extrachromosomal arrays (*Jans et al., 2009*; *Csankovszki et al., 2004*). To address this contradiction, we performed a series of ectopic *rex* insertions on chromosome-II (*Figure 5A*) with the following two predictions.

First, if *rex* sites are mere barriers for condensin DC, then increasing the number of ectopic *rex* sites would not affect the mean DPY-27 ChIP-seq signal on chromosome-II. In other words, the observed binding at or near *rex* sites would be a consequence of redistribution of binding events on chromosome-II. Since the frequency of binding would be dictated by the length of chromosome-II, the mean signal on chromosome-II should remain unaffected by insertion of *rex* sites. Contrary to this prediction,

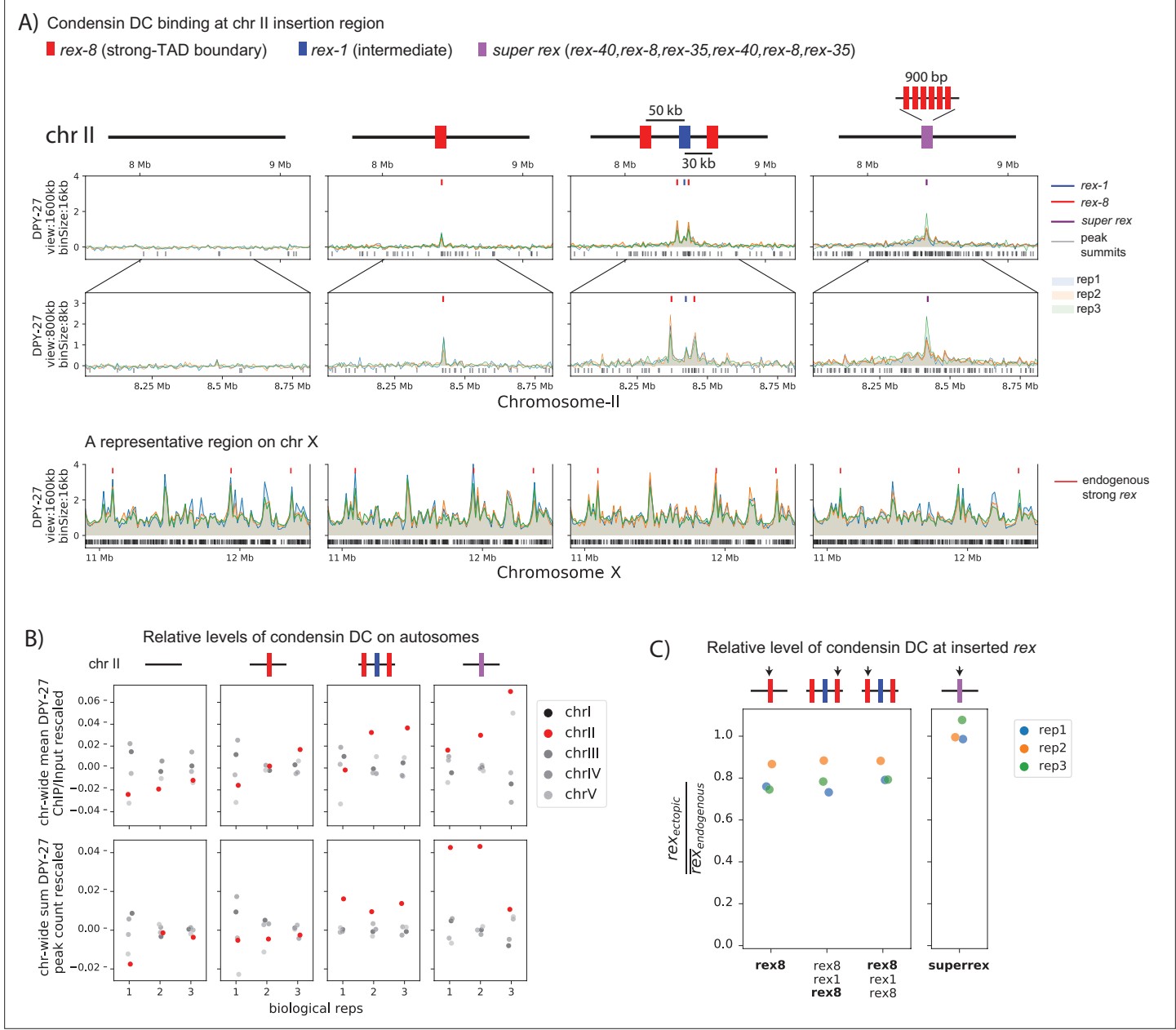

**Figure 5.** Condensin dosage compensation (DC) is recruited to and spreads in either direction from the ectopically inserted *recruitment elements on the X-chromosome* (*rex*) sites on chromosome-II in L3. (**A**) Snapshot of a region on chromosome-II where *rex* sites are inserted. Shown are four different conditions in L3 stage: wild-type, ERC61 (single *rex-8* insertion) (**Albritton et al., 2017**), ERC63 (*rex-8, rex-1, rex-8*) [33], and ERC90 *super rex*, an array of six truncated, midsection (150 bp) of three strong *rex*'s: *rex-40, rex-8, rex-35*, repeated. Binding surrounding the insertion sites in both directions increases with increased number and strength of inserted *rex*es. The levels of condensin DC ectopically recruited to chromosome-II remain weaker than the endogenous binding at X-chromosome (lower panel). (**B**) The enrichment of condensin DC on each autosome plotted as chromosome-wide mean enrichment and total number of binding peaks calculated from DPY-27 ChIP-seq data. The ChIP-seq values or the total number of peaks are rescaled for each data, such that the mean of autosomes excluding chromosome-II is fixed to 0 and that of X-chromosome is at 1. The plot highlights weak but reproducible recruitment of more condensin DC to chromosome-II by *rex* insertion. (**C**) The level of condensin DC binding at the inserted strong *rex-8* (indicated by arrow in each strain) is comparable to the endogenous, as indicated by the ChIP-seq signal at inserted *rex* site (400 bp) normalized to the mean of all endogenous strong *rex* sites on the X.

The online version of this article includes the following figure supplement(s) for figure 5:

**Figure supplement 1.** Condensin dosage compensation (DC) is recruited to and spreads in either direction from the ectopically inserted *recruitment elements on the X-chromosome* (*rex*) sites on chromosome-II in embryo.

**Figure supplement 2.** mRNA-seq data comparing *recruitment elements on the X-chromosome* (*rex*) insertion strains to wildtype.

increasing the number of ectopic *rex* sites increases the mean DPY-27 ChIP signal and the number of peaks on chromosome-II in both L3 (*Figure 5B*) and embryos (*Figure 5—figure supplement 1B*).

Second, if *rex* sites are mere barriers for condensin DC, then we expect a decrease in the DPY-27 ChIP-seq signal at *rex* sites with subsequent insertions. That is, redistribution of condensin DC from non-*rex* sites to *rex* implies that the additional *rex* sites would effectively function as barriers to each other. Therefore, addition of another *rex* site in close proximity would dampen the signal at the initially inserted *rex* site due to multiple *rex* sites competing for the access to incoming condensin DC in cis. Again, contrary to this prediction, we observe that the presence of two flanking bidirectional barriers, *rex-8*, enhances binding at *rex-1* (*Figure 5—figure supplement 1C*) and minimally affects binding at *rex-8* (*Figure 5C*, *Figure 5—figure supplement 1C*). Importantly, ChIP-seq binding levels here tightly follow the results of quantitative ChIP experiments performed in the three *rex* insertion embryos (*Albritton et al., 2017*). Therefore, *rex* sites behave more cooperatively than competitively as observed by recruitment at the *rex* sites.

In summary, the extrachromosomal recruitment assays and ChIP binding data in ectopic insertions cannot be sufficiently explained by a 'barrier-only' function for *rex* sites. These observations are more consistent with *rex* sites being bona fide loading sites for condensin DC. Furthermore, we observe ectopic recruitment and spreading as a qualitative function of 'strength' and 'number' of the *rex* sites. Insertion of a weaker *rex* (*rex-1*) results in no recruitment or spreading while insertion of a single strong *rex* (*rex-8*) leads to recruitment without spreading. Insertion of combination of *rex-1* and *rex-8* leads to both recruitment and spreading (*Figure 5A*, *Figure 5—figure supplement 1A*), suggesting the of *rex* sites also behave cooperatively in regards to spreading from the *rex* sites.

To further test the correlation between the *rex* element 'strength' and the level of condensin DC recruitment and spreading, we made a '*super rex*' element combining six strong 150 bp rex sequences into one 900 bp element and inserting into chromosome-II at the same location as the single strong *rex-8* insertion. Insertion of the *super rex* indeed was sufficient to recruit condensin DC as a point source and led to spreading of condensin DC more than that of the single *rex-8* (*Figure 5A*). Therefore, we conclude that *rex* sites are loading sites for condensin DC in addition to being bidirectional barriers.

## Ectopic recruitment of condensin DC to chromosome II achieves weak binding and domain-level repression

Condensin DC represses transcription initiation by a factor of ~2 across the entire X-chromosomes (*Kruesi et al., 2013*; *Kramer et al., 2016*; *Kramer et al., 2015*). The level of condensin DC binding is proportional to repression as seen in X;A fusion chromosomes (*Street et al., 2019*). In ectopic recruitment experiments, the level of condensin DC binding in the autosomes is lower than that of the X (*Figure 5A*). We performed mRNA-seq in two *rex* insertion strains and observed that genes were not individually repressed by condensin DC, but there was a measurable effect when changes were analyzed at the domain level within ~200 and ~500 kb bins and even at the chromosomal level (*Figure 5—figure supplement 2*). This supports the idea that condensin DC induces domain-wide repression whose effect size is proportional to the level of condensin DC binding.

## Ectopic insertion of *rex* elements leads to condensin DC spreading and formation of loop-anchored TADs

To understand if inserted *rex* elements can sufficiently form TADs, we performed Hi-C in three notable experimental conditions based on ChIP-seq data in *Figure 5A*.

First, the single *rex-8* insertion, where there is recruitment with minimal spreading. In the bacterial *parS*/parB system, condensins not only traverse through but also collide with each other (*Brandão et al., 2021*). We hypothesized that if the loading of condensin DC functions as a barrier for another SMC complex, such as cohesin, then recruitment alone would coincide with boundary formation. Insertion of a single *rex-8* created a TAD boundary on the X (*Figure 3*) but not on chromosome II (*Figure 6*). Therefore, inserted *rex-8* did not serve as an indiscriminate loop extrusion barrier for other SMC complexes. This is in agreement with previous work showing that insertion of strong *rex* sites (*rex-47*, *rex-8*, or *rex-14*) on chromosome-I does not result in boundary formation (*Anderson et al., 2019*).

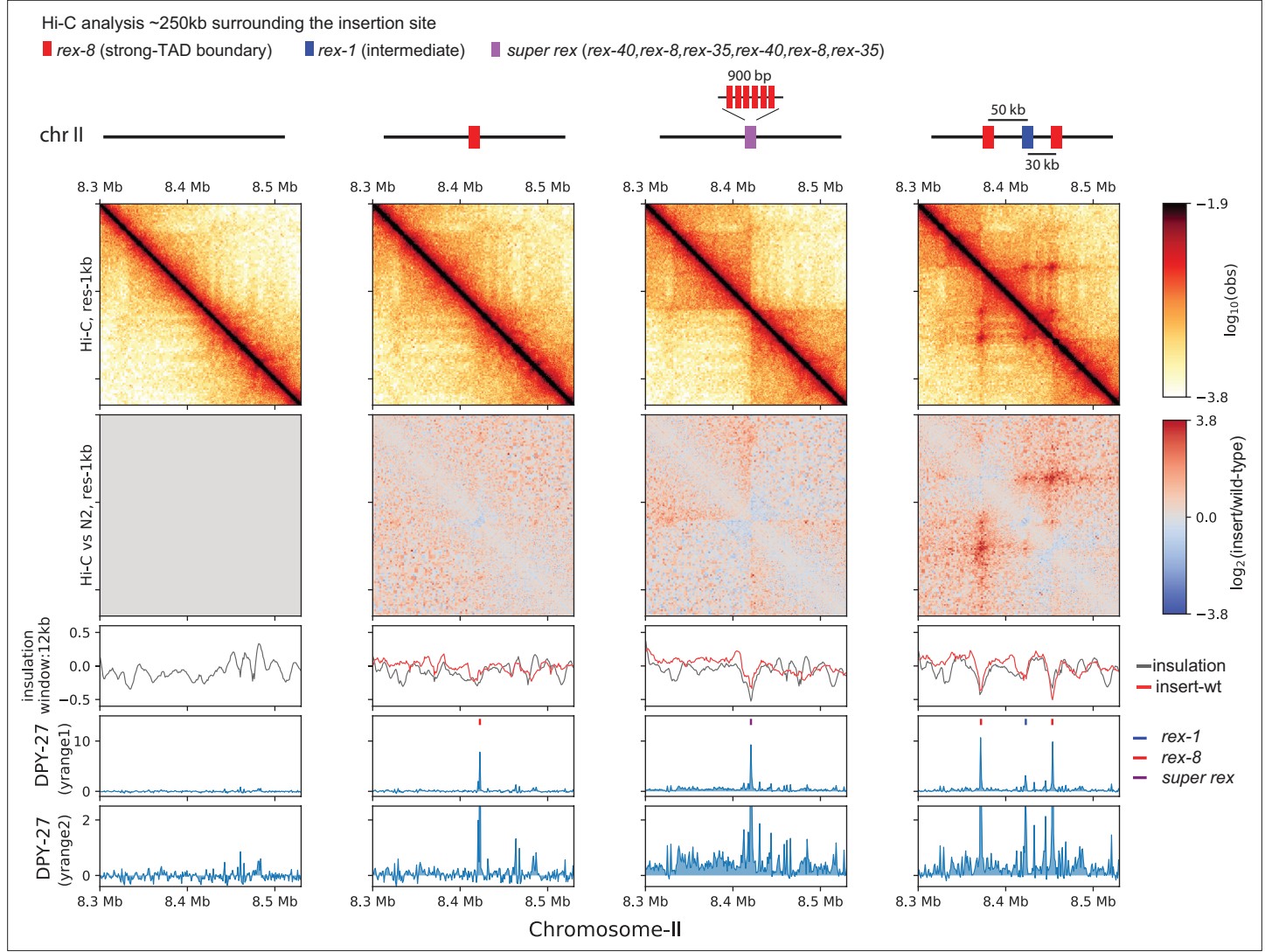

**Figure 6.** Ectopic insertion of *rex* elements that lead to condensin dosage compensation (DC) spreading form loop-anchored topologically associating domains (TADs). Snapshot of a region on chromosome-II where *recruitment elements on the X-chromosome* (*rex*) sites are inserted. Shown are four different conditions in L3 stage: wild-type, single *rex-8* insertion at one site, six concatenated strong *rex* (*super rex*) insertion at the same site and insertion of *rex-8*, *rex-1*, and *rex-8 at a distance from each other spanning ~80* kb. Two different y-range cutoffs for ChIP-seq data are provided to highlight the level of condensin DC spreading in each strain. Log2ratio Hi-C plots or subtraction of insulation score (insertion-wild type) plots are generated using wild-type data mapped to the corresponding insertion genome in comparison.

The online version of this article includes the following figure supplement(s) for figure 6:

**Figure supplement 1.** Spreading coincides with changes in 3D contact.

**Figure supplement 2.** Control regions for ectopic *recruitment elements on the X-chromosome* (*rex*) insertion experiment.

Second, the *super rex* insertion, where there are both recruitment and spreading in either direction from a singular source. In the bacterial *parS*/parB system, condensin is recruited to and spreads from the *parS* site to form a secondary diagonal on Hi-C matrix (**Stigler et al., 2016**; **Sullivan et al., 2009**; **Wang et al., 2017**; **Wang et al., 2018**). This pattern suggests an effective two-sided loop extrusion initiating from *parS* site. Alternatively, in simulation, one-sided loop extrusion from a loading site results in 'flames' (also referred to as 'stripes'). This is observed as a '+' centered at the loading site (**Banigan et al., 2020**). Consistent with one-sided extrusion, we observe flames emanating from *super rex* (**Figure 6**). The flames off the endogenous strong *rex* sites are also apparent on the X-chromosome (**Figure 1C**). Additionally, we observe an insulation effect at the *super rex*. Given that *rex* sites

do not function as indiscriminate barriers for other SMC complexes, the insulation effect implies that condensin DC also facilitates contacts between non-*rex* elements on the same side of the *rex* site.

Third, the three *rex* insertion, where there are both recruitment and spreading from multiple sources. In the three-*rex* insertion, we observe all features of TADs, including enriched square contacts, insulation, flame, and corner peak (*Figure 6*). This suggests that the presence of multiple *rex* sites can sufficiently recapitulate TADs observed on X-chromosome. A flame that extends past the right *rex-8* implies that a subset of one-sided LEFs loaded on the left *rex-8* can move past the right *rex-8*, suggesting a probabilistic barrier function by the *rex* sites.

When zoomed out, the spreading of condensin DC observed in *super rex* and three *rex* insertions coincides with the local increase in the insulation score compared to the wildtype (*Figure 6—figure supplement 1*, red arrows). This suggests that observed changes in 3D structure are not simply a redistribution of existing contact frequencies but rather a formation of additional contacts. Such effects are not observed in the control region on chromosome-I (*Figure 6—figure supplement 2*). In summary, the facilitation of 3D contacts that coincides with the spreading of condensin DC along with the relative insulation at the ectopic *rex* sites supports a model based on loop extrusion by condensin DC.

## Discussion

### The cooperativity of *rex* sites contributes to the X-specific recruitment and spreading of condensin DC

In previous work, we proposed that cooperation between *rex* sites ensures X-specific and robust binding of condensin DC to an entire chromosome (*Albritton and Ercan, 2018*). Our ChIP-seq analysis of spreading in ectopic insertion strains suggests that the cooperativity is also involved in spreading. The insertion of a single strong *rex-8* on chromosome-II resulted in recruitment but not spreading of condensin DC. This lack of spreading was 'rescued' by the use of engineered *super rex* (an array of six strong *rex* sites) or insertion of additional *rex* sites (*rex-1* and *rex-8*), suggesting that the cooperative function of *rex* sites is reflected in both recruitment and spreading.

In this study, insertion of three *rex* sites in close proximity (within 100 kb) recapitulated the Hi-C features of the loop-anchored TADs on the X-chromosomes. A previous attempt of three *rex* insertions spaced far apart (within 3 MB) resulted in weak recruitment of DCC and no changes on Hi-C matrix (*Anderson et al., 2019*). Therefore, cooperativity must be distance dependent. On the autosome, the inserted *rex* sites may have to compete against the X. Indeed, even with a *super rex* insertion that recruited at a level comparable *rex* sites on the X, the level of spreading remained lower than that of the endogenous X-chromosome. On the X, activity of many weaker *rex* sites in between the 1 MB distance (average between strong *rex* sites) may contribute to the robust recruitment and spreading of condensin DC.

### Previous models of condensin DC binding on the X-chromosomes

Currently, there are two seemingly contradictory models for how condensin DC binding occurs on the X-chromosome. The first one is the recruitment to and spreading from *rex* sites some of which include a 12 bp sequence motif required for their function (*Albritton and Ercan, 2018*; *Meyer, 2022a*; *Meyer, 2022b*). It is not clear what protein binds directly to the motif, but the candidates are the SDC proteins required for condensin recruitment to the X. Unlike *rex* sites, many other segments of the X-chromosomes cannot autonomously recruit condensin DC (*Jans et al., 2009*; *Csankovszki et al., 2004*). Thus, condensin DC must 'spread' from *rex* to other sites on the X-chromosomes. Such 'spreading' phenomenon was later observed on X;A fusion chromosomes (*Ercan et al., 2009*; *Pferdehirt et al., 2011*).

The second model is the uniform loading and stalling of condensin DC at *rex* sites (*Anderson et al., 2019*; *Meyer, 2022a*; *Meyer, 2022b*). This model is derived from the cohesin-related experiments in the mammalian system (*Fudenberg, 2017*; *Kim et al., 2019*; *Golfier et al., 2020*). If *rex* sites were preferential loading sites for a two-sided LEF, one would observe a secondary perpendicular diagonal which is not present in the Hi-C data. It was thus proposed that condensin DC loads uniformly across the X-chromosome (*Anderson et al., 2019*; *Meyer, 2022a*; *Meyer, 2022b*). The strength of this model is in its ability to explain TADs observed on the X-chromosome using minimalistic properties. However, unlike the mammalian cohesin, condensin DC lacks in vitro evidence showing that it is a two-sided LEF. Additionally, this model necessitates an added complexity in order to explain how

condensin DC is specifically targeted to the X-chromosome. In the following section, we reconcile the two models by proposing that the loading of condensin DC at *rex* sites can result in TAD formation.

## A model to explain X-specific recruitment of condensin DC and formation of loop-anchored TADs by *rex* sites

We consider three non-mutually exclusive features of the DCC system each with increasing complexity in order to explain how *rex* sites give rise to TADs observed on X-chromosome (*Figure 7*).

1) One-sided loop extrusion from *rex* sites: We observe that *rex* sites are both loading sites acting as the source of condensin DC spreading and bidirectional barriers. We begin by proposing that condensin DC is a one-sided LEF based on the presence of flames and the absence of a perpendicular diagonal. In simulation, one-sided LEFs cannot form corner-peaks due to high frequency of gaps between two oppositely oriented LEFs (*Banigan et al., 2020*). This is unlikely to be a limitation in our system, because (i) in simulation, the corner-peaks can be 'rescued' with strong preferential loading at the boundary and (ii) a recent preprint reported that the cleavage of DPY-26, a kleisin subunit shared by condensin I and DC, did not eliminate *rex-rex* corner-peaks (*Das, 2022*). Therefore, the corner-peak does not necessitate an explanation based on loop extrusion.

The main limitation of condensin DC loading and extruding only from the *rex* sites is the inability to explain condensin-DC-mediated contacts within TADs (i.e. form transient loops between two non-*rex* elements). While it is possible that *rex* sites are merely preferential loading sites, how non-*rex* loading evades autosomes raises more questions. One explanation is that such off-target loading is skewed toward the X-chromosome by the increased condensin DC 'local concentration' produced by SDCs. However, the loss of SDC-2 does not result in global redistribution of condensin DC; in the absence of SDC-2, the fluorescence recovery after photobleaching (FRAP) recovery of condensin DC shows similar dynamics to that of free-floating molecules (*Breimann et al., 2022*). Therefore, the simplest conclusion reconciling many observations is that the *rex* sites act as condensin DC loading sites, and additional features of the DCC system explain the non-*rex* loops within TADs.

2) Loader spreading from the *rex* sites: It has been proposed that the condensin DC loaders also 'spread,' although to a lesser extent than condensin DC, leading to SDC-2, SDC-3, and DPY-30 enrichment at active promoters on the X-chromosome (*Ercan et al., 2007*; *Albritton et al., 2017*; *Street et al., 2019*; *Pferdehirt et al., 2011*). One possible model is that these SDC-bound promoters function as secondary loading sites. However, as can be seen from *Figure 1B* and previous peak analysis (*Albritton et al., 2017*), the number of SDC-2 (the hermaphrodite specific protein that initiates condensin DC recruitment to the X) binding sites on X-chromosome is an order of magnitude below that of condensin DC subunits. In other words, SDC-2 distribution may contribute but is too sparse to mimic uniform loading across X-chromosome, a necessary feature in order to enrich contacts within a TAD.

3) Loop-anchor displacement from the *rex* sites: We propose that upon loading, condensin DC may begin one-sided loop extrusion, but the inactive loop-anchor at the *rex* site is prone to being displaced. This model is inspired from the simulation work by *Banigan et al., 2020*, which informs that one-sided loop extrusion from a loading site whose inactive anchor is prone to displacement by diffusion (semi-diffusive one-sided loop extrusion (LE) model with a loading site) results in an increasingly murkier flame with increasing rate of diffusion (*Banigan et al., 2020*). Movement of the inactive anchor would allow condensin DC to form transient loops between two non-*rex* elements forming increased contacts within TADs. It is possible that the anchor displacement is promoted by another SMC complex, subsequent loading of condensin DC, intermittent translocation without loop formation, or diffusion.

A candidate SMC complex displacing condensin DC loop anchor is cohesin, whose loader/processivity factor PQN-85 (NIPBL homolog in *C. elegans*) is enriched at the *rex* sites (*Kranz et al., 2013*). The hypothesis that incoming condensin DC molecules leading to anchor displacement is based on the idea that a one-sided LEF uses the loading site as loop extrusion anchor. This puts the currently extruding and free-floating LEFs at a competition for the loading site. One possible outcome of this competition is that the subsequent loading event displaces the inactive anchor of currently extruding LEF, which is then further pushed away from the loading site by the initiation of loop extrusion by newly loaded LEF. Anchor displacement could also happen through ATP-dependent directional translocation without loop formation as observed in double tethered assay where DNA is tightly anchored

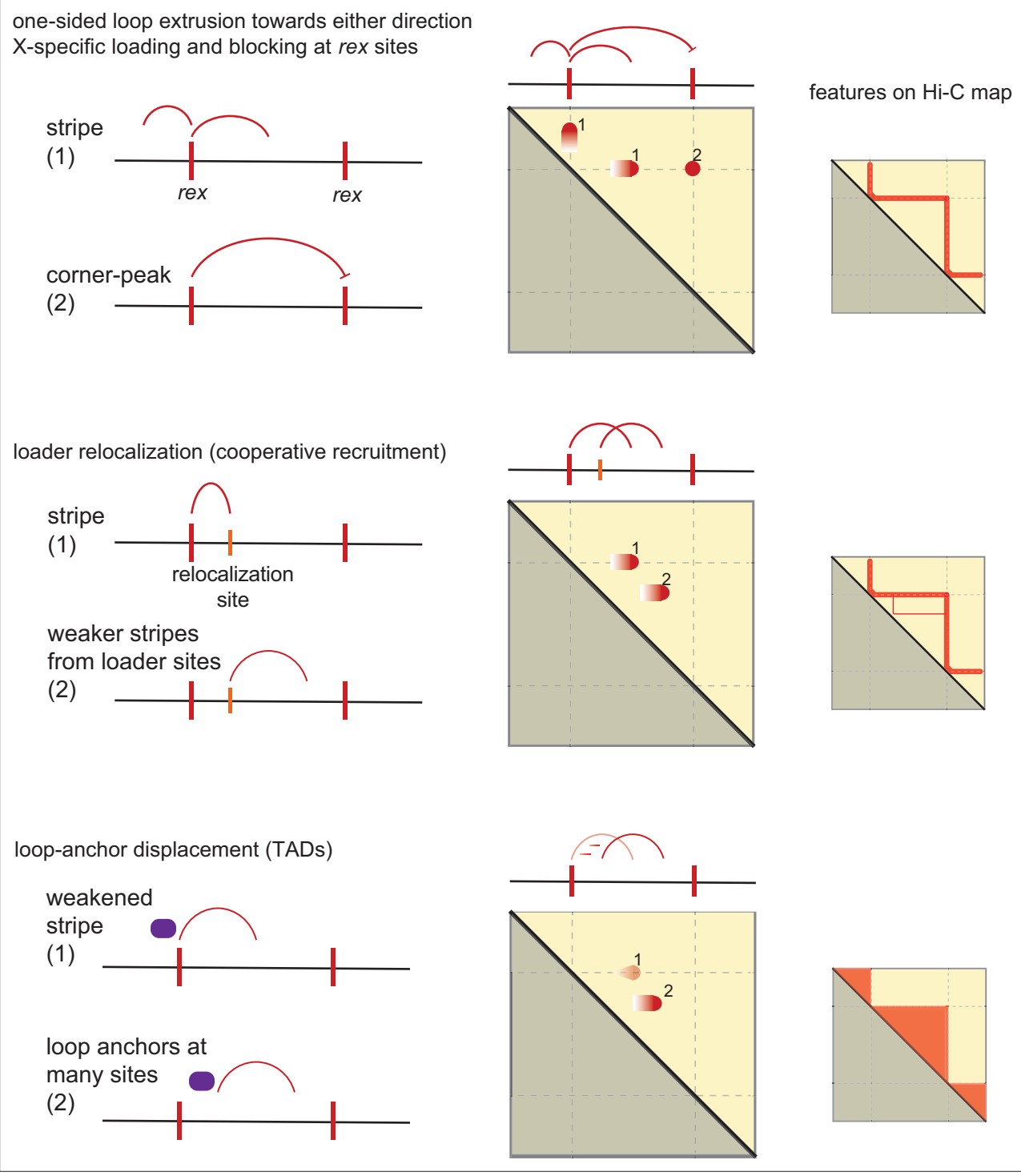

**Figure 7.** Model of condensin dosage compensation (DC) recruitment and spreading forming loop-anchored topologically associating domains (TADs). Shown are three conceptually distinct features that contribute to TAD formation in *C. elegans* dosage compensation complex (DCC) system. The model as a group contains the following properties: (i) *rex* is a bidirectional barrier, (ii) *rex* is a loading site, (iii) *rex* can activate/enhance weaker 'secondary loading sites,' (iv) condensin DC is a one-sided loop extruder, and (v) condensin DC's loop-anchor is prone to displacement.

at two ends (*Terakawa et al., 2017*). This observation was explained by combining loop-capture and power-stroke model, in which the mode of translocation converts from loop extrusion to linear translocation with increasing DNA tension, suggesting that the two different modes of active translocation, LE and non-LE, can be embedded in the same SMC complex (*Nomidis et al., 2022*).

### Establishment and maintenance of the X-chromosome 3D structure

In this paper, we used genome editing and performed ChIP-seq and Hi-C to analyze the binding and function of condensin DC. The limitation of our study is the lack of a temporal component, addressing how the structure of the X-chromosome is initialized and maintained. Previous work showed that DCC localizes to and represses X-chromosome in the early stages of embryo development (~40 cells) (*Kramer et al., 2015*; *Dawes et al., 1999*). In mixed stage embryos and L3 worms, we may be analyzing a maintenance/equilibrium state of X-chromosome organization. Hi-C data in a recent preprint presented that upon degradation of SDC-3 or cleavage of kleisin subunit DPY-26, TADs disappear but the *rex-rex* corner peaks remain unperturbed (*Das, 2022*). This is in contrast to the genetic loss of SDC-2 resulting in the complete loss of all features of TADs (*Crane et al., 2015*; *Anderson et al., 2019*). It is possible that *rex-rex* interactions initiated by SDC-2 mediated recruitment and condensin DC loop extrusion in early embryogenesis are maintained by a different mechanism during later development.

### The role of condensin DC loop extrusion and TADs in transcription repression of the X-chromosomes

Previously, we showed that genes located at the autosomal spreading region in the X;V fusion chromosome are repressed due to condensin DC spreading (*Street et al., 2019*). Here, we show that this repression does not coincide with TAD formation on the chromosome-V side, and despite a clear loop-anchored TAD in three *rex* insertions, there was minimal effect on transcription. One hypothesis is that the process of loop extrusion by condensin DC is necessary for repression rather than insulation of 3D contacts by *rex* sites. Consistent with this idea, deletion of the eight strong *rex* sites did not result in changes in Hi-C distance decay curve, implying that the loop extrusion activity of condensin DC is largely unperturbed (*Anderson et al., 2019*). Similarly, shortening of 3D contacts indicative of condensin DC loop extrusion in the X;V fusion chromosome coincides with repression. Alternatively, condensin DC loop extrusion is a mechanism evolved to distribute the complex across long distances from the *rex* sites to accomplish chromosome-wide dosage compensation. Consistent with this idea, upon TOP-2 depletion, the processivity of condensin DC loop extrusion reduces, and the spreading of the complex to the chromosome also reduces (*Morao et al., 2021*). The repressive effect of condensin DC may be local and loop extrusion independent. In agreement with this, a null mutant of *dpy-21* (a non condensin DCC subunit) results in X upregulation without significantly altering Hi-C interactions (*Breimann et al., 2022*). Future work separating Hi-C features and DC should shed light on how a condensin complex was co-opted to repress transcription of an entire chromosome.

## Materials and methods

All sequencing data are available at Gene Expression Omnibus under GSE168803.

### Strains

Unless otherwise noted, strains were grown on nematode growth medium (NGM) media under standard worm culturing conditions. N2 wild type, one *rex* insertion ERC06 (knuSi254[SNP400bprex-1, unc-119(+)] II; unc-119[ed3] III), two *rex* insertion ERC62 (ersIs26[X:11093923-11094322(rex-8)], II: 8449965); knuSi254[SNP400bprex-1, unc-119(+)] II; unc-119[ed3] III, and three *rex* insertion ERC63 (ersIs27[X:11093923-11094322(rex-8)], II:8371600, II:8449968); (knuSi254[SNP400bprex-1, unc-119(+)] II; unc-119[ed3] III) strains were previously described (*Albritton et al., 2017*). *Super rex* insertion ERC90 (ersIs62[superrex(rex40,rex8,rex35)x2],II:8420106). Upstream oriented rex-8 insertion strain is ERC69 (ersIs33[X:11093924-11094281(rex-8)], X:14373128]), downstream oriented *rex* insertion ERC80 (ersIs52[X:11094281-11093924[rex-8reverse], X:14373128]). X-V fusion YTP47 (XR-VR) and condensin DC spreading was described in *Ercan et al., 2009*. Primer sequences used in the generation of the CRISPR strains are included in *Supplementary file 1*.

### Constructs and transgenes

dCas9-Suntag targeting strain containing the sgRNA is JZ2005 with the genotype pySi27(Pfib-1:NLS::scFv::sfGFP::NLS::tbb-2 3'UTR/unc-119[+]) I; pySi26(Pfib-1::NLS::dCas9::24xGCN4::NLS::tbb-2 3'UTR/unc-119[+]) II; unc-119(ed3)/+III; pyIs1002(pU6::sgRNA- X227/Punc-122::mCherry), and without

the sgRNA is JZ1973 (pySi[Pfib-1:NLS:scFv:sfGFP/unc-119(+)]) I; pySi(Pfib-1:NLS:dCas9:24xGCN4/unc-119[+] II; unc-119[ed3] III).

pNLZ10 (Pfib-1::NLS::dCas9::24xGCN4::NLS::tbb-2 3'UTR) construct contains pCFJ150 vector backbone (Addgene plasmid # 19329). *SV40* NLS::dCas9::*egl-13* NLS:: tbb-2 3'UTR was derived from pJW1231 (Phsp-16.48::NLS::dCas9::EGFP::NLS::tbb-2 3'UTR; a generous gift from Dr. Jordan D Ward), which was made by introducing D10A and H840A mutations into pMB66 (Phsp-16.48:NLS:Cas9:EGFP:NLS:tbb-2 3'UTR; *Waaijers et al., 2013*). To produce catalytically dead Cas9 (dCas9) and then subcloned into pCFJ150 vector. NLS: nuclear localization signal. A codon-optimized 6.25 copies of GCN4 fragment (*Tanenbaum et al., 2014*) were synthesized by Integrated DNA Technologies (IDT), and another three multiple GCN4 fragments containing artificial introns were made by PCR amplification to generate 24 copies of GCN4.

pNLZ11 (Pfib-1::NLS::scFv::sfGFP::NLS::tbb-2 3'UTR) construct has the pCFJ210 vector backbone (Addgene plasmid # 19329). A codon-optimized scFv::sfGFP fragment (*Tanenbaum et al., 2014*) was ordered from IDT. *fib-1* promoter was PCR amplified from worm genomic DNA. *tbb-2* 3'UTR was amplified from pJW1231 (Phsp-16.48::NLS::dCas9::EGFP::NLS::tbb-2 3'UTR). *SV40* and *egl-13* NLS sequences were added with PCR primers used for amplifying assembly fragments.

pBHC1131 (PU6::sgRNA-X227) construct (a generous gift from Baohui Chen) was derived from pDD162 (Addgene plasmid #47549) (*Dickinson et al., 2013*) and targets an X-chromosome repetitive region with guide RNA sequence 5'-GGCGCCCATTTAAGGGTA-3'. The sgRNA construct was modified with optimized sgRNA scaffold (*F+E*) which can improve the CRISPR imaging efficiency in human cells (*Chen et al., 2013*).

Pfib-1::NLS::dCas9::24xGCN4::NLS::tbb-2 3'UTR and Pfib-1::NLS::scFv::sfGFP::NLS::tbb-2 3'UTR were single-copy inserted into worm genome by MosSCI using direct injection protocol (*Frøkjaer-Jensen et al., 2008*). PU6::sgRNA-X227 was injected at 200 ng/µL concentration with P*unc-122*::mCherry as the co-injection marker to get a transgenic extrachromosomal array line, which was subsequently integrated into worm genome by trimethylpsoralen ultraviolet (TMP-UV) method.

## mRNA-seq and ChIP-seq

ChIP-seq and mRNA-seq experiments were performed as previously described (*Albritton et al., 2017*). Antibody information and new and published datasets used are given in *Supplementary file 1*. We aligned 50–75 bp single-end ChIP-seq reads to *C. elegans* genome version WS220 or WS220 variants for ChIP-seq performed in insertion strain or X-V fusion strain using bowtie2 2.3.2 with default parameters (*Langmead and Salzberg, 2012*). Bam files were then sorted and indexed using samtools version 2.1.1 (*Ramirez-Gonzalez et al., 2012*). ChIP enrichment was normalized by dividing to input using DeepTools bamCompare using the following parameters: CPM, bin-size of 10 bp, ignore duplicates, extend reads to 200 bp, exclude chrM, MAPQ 1, and remove blacklisted regions (*Amemiya et al., 2019*). MACS2 version 2.1.1 (https://github.com/macs3-project/MACS; *Liu, 2022*) was used for fragment size prediction and for peak calling. Peaks are called using individual replicates and combined bam files with minimum false discovery rate of 0.05. Bedtools intersect was used to determine overlapping peaks between replicates and merged bam files; only those present in the majority of the replicates were chosen as the final peaks to be plotted for visualization in *Figure 4*. For all DPY-27 ChIP-seq data, input normalized signal at each 10 bp genomic bin (xi) is rescaled (xf) using the following method:

$x_f = (x_i - A)/(X - A)$, where A is the mean of autosomes excluding chrII, and X is the mean of chrX.

The final plotted data has the mean of autosomes (excluding chromosome-II) of 0 and the mean of chromosome-X of 1. The rationale for this rescaling is that DPY-27 ChIP-seq experiment has a technical variability with regards to the enrichment of reads across the chromosome-X. If ChIP-seq 'works well,' proportionally more reads are mapped to the X-chromosome, which is necessarily coupled with proportionally less reads mapping to autosomes. This variability makes ChIP-seq hard to compare across conditions. Therefore, we use total signal on autosomes (excluding chromosome-II) and that of X-chromosome as biologically reliable data points. The chromosome-II was excluded because chromosome-II is the 'experimental' chromosome where *rex* sites are inserted thus the signal may differ between conditions. This makes chromosome-II not a reliable biological data point across experimental conditions. For chip-chip data from Ercan 2009 used in *Figure 2*, we use the distal (farthest away from the fusion site) 5 MB of chrX side as X and distal 5 MB of chrV side as A. For SDC-3

ChIP-seq data, input normalized signal is simply autosome centered since $A$ and $X$ have a difference of near 0. For any other ChIP-seq data used in the paper, simply input normalized data is used.

## Hi-C

Worms were grown on standard NGM plates, and gravid adults were bleached to obtain embryos, which were crosslinked with 2% formaldehyde and stored at –80°C (*Ercan et al., 2007*). Frozen embryos were then resuspended in and crosslinked with 2% formaldehyde in M9 for another 30 min. The embryos were spun down at 6000 g for 30 s and washed once with 100 mM Tris Cl pH 7.5 and twice with M9. The embryo pellet was resuspended in 1 mL embryo buffer (110 mM NaCl, 40 mM KCl, 2 mM CaCl2, 2 mM MgCl2, and 25 mM HEPES-KOH pH 7.5) containing one unit chitinase (Sigma), digested approximately 15 min, and blastomeres were washed with embryo buffer twice by spinning at 1000 g 5 min. The pellet was resuspended in Nuclei Buffer A (15 mM Tris–HCl pH 7.5, 2 mM MgCl2, 0.34 M Sucrose, 0.15 mM Spermine, 0.5 mM Spermidine, 1 mM DTT, 0.5 mM PMSF [1× Calbiochem Protease Inhibitor cocktail I], 0.25% NP-40, 0.1% Triton X-100), centrifuged at 1000 g for 5 min at 4°C then resuspended in 1.5 mL Nuclei Buffer A. The embryos were then dounced 10× with a loose pestle and 10× with a tight pestle. The nuclei were separated from the cellular debris through spinning down the dounced material at 200 G, then collecting the supernatant containing the nuclei into a separate tube. The pellet was resuspended in 1.5 mL, and the douncing process was repeated four times. Each individual supernatant containing nuclei was checked for quality by DAPI staining, and those without debris were pooled and spun down at 1000 G for 10 min at 4°C. Approximately 20 µL nuclei pellet were used to proceed to Arima Hi-C per the manufacturer's instructions. Library preparation was performed using the KAPA Hyper Prep Kit using the protocol provided by Arima. Paired-end Illumina sequencing was performed with Nextseq or Novaseq.

## Hi-C data processing and analysis

Hi-C data analysis: The Hi-C data was mapped to ce10 (WS220) reference genome using default parameters of the Juicer pipeline version 1.5.7 (*Durand et al., 2016*). The biological replicates were combined using juicer's mega.sh script. The mapping statistics from the inter_30.txt output file are provided in *Supplementary file 1*. The inter_30.hic outputs were converted to cool format using the hicConvertFormat of HiCExplorer version 3.6 (*Wolff et al., 2018*; *Ramírez et al., 2018*) in two steps using the following parameters: (1) `--inputFormat hic, --outputFormat cool` and (2) `--inputFormat cool --outputFormat cool --load_raw_values`. The cool file was balanced using cooler version 0.8.11 using the parameters: `--max-iters 500`, `--mad-max 5`, and `--ignore-diags 2` (*Abdennur and Mirny, 2020*). The balanced cool file was used for all downstream analysis. For computing log-binned P(s), its log-derivative, insulation scores, and on-diagonal pile-up analysis at 17 strong *rex* sites (*Albritton et al., 2017*), cooltools version 0.4.0 (https://github.com/open2c/cooltools; *Fudenberg, 2022*) was used. For visualizing ChIP-seq data with Hi-C data in python, pyBigwig version 0.3.18 (https://github.com/deeptools/pyBigWig; *Ryan, 2021*) was used. The jupyter notebook containing computational workflow for each figure is publicly available (https://github.com/ercanlab/2022_Kim_Jimenez_et_al, copy archived at swh:1:rev:9b46ddadb4f296f8f238e06dc66b-b4e9a58b21a2; *Kim, 2022*).ChIP-seq mapping statistics, Hi-C juicer statistics, and RNA-seq TPM values for each replicate is provided. The description of strains and the primers used to generate the strains are also provided.

## Acknowledgements

SE and research in this manuscript were supported by the National Institute of General Medical Sciences of the National Institutes of Health under award number R35 GM130311. DJ, JK, and MK were supported in part by NIGMS Predoctoral Fellowship T32HD007520. We thank Dr. Noelle L'etoile, who generously supported Bo Zhang's postdoctoral work that included dCas9 targeting, with grants NIBIB R33 EB019784 and NIDCD R01 DC005991. We thank Jordan Ward and Baohui Chen for plasmids. We thank Gencore at the NYU Center for Genomics and Systems Biology. We also thank Shaun Mahony, Lila Rieber, Pedro Rocha, Ramya Raviram and Jane Skok who helped with experimental trials prior to adopting Hi-C, and Arima genomics for technical support.

## Additional information

### Funding

| Funder | Grant reference number | Author |
|---|---|---|
| National Institute of General Medical Sciences | R35 GM130311 | Sevinc Ercan |
| National Institute of General Medical Sciences | T32HD007520 | Jun Kim |
| National Institute of Biomedical Imaging and Bioengineering | R33EB019784 | Bo Zhang |
| National Institute on Deafness and Other Communication Disorders | R01DC005991 | Bo Zhang |

The funders had no role in study design, data collection and interpretation, or the decision to submit the work for publication.

### Author contributions

Jun Kim, Formal analysis, Investigation, Visualization, Writing – review and editing; David S Jimenez, Conceptualization, Formal analysis, Investigation, Visualization, Writing – original draft, Writing – review and editing; Bhavana Ragipani, Lena A Street, Maxwell Kramer, Sarah E Albritton, Investigation, Methodology; Bo Zhang, Resources, Investigation, Methodology; Lara H Winterkorn, Ana K Morao, Investigation; Sevinc Ercan, Conceptualization, Resources, Supervision, Funding acquisition, Visualization, Writing – original draft, Writing – review and editing

### Author ORCIDs

Jun Kim ⓘ http://orcid.org/0000-0001-9473-3093
Sevinc Ercan ⓘ http://orcid.org/0000-0001-7297-1648

### Decision letter and Author response

Decision letter https://doi.org/10.7554/eLife.68745.sa1
Author response https://doi.org/10.7554/eLife.68745.sa2

## Additional files

### Supplementary files

• Supplementary file 1. Data summary and statistics. ChIP-seq mapping statistics, Hi-C juicer statistics, and RNA-seq Transcript per million (TPM) values for each replicate are provided. The description of strains and the primers used to generate the strains are also provided.

• Transparent reporting form

### Data availability

Sequencing data have been deposited in GEO under GSE168803.

The following dataset was generated:

| Author(s) | Year | Dataset title | Dataset URL | Database and Identifier |
|---|---|---|---|---|
| Kim J, Jimenez DS | 2021 | Condensin DC spreads linearly and bidirectionally from recruitment sites to create loop-anchored TADs in *C. elegans* | https://www.ncbi.nlm.nih.gov/geo/query/acc.cgi?acc=GSE168803 | NCBI Gene Expression Omnibus, GSE168803 |

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
