## [Editor Report]

This paper is likely to be of broad interest to researchers in the chromosome biology field. With specific loading sequences identified, the Condensin dosage compensation complex studied here provides an elegant system to investigate the in vivo activities of SMC complexes. Combining Hi-C, ChIP-seq and RNA-seq, the authors reveal that the complex spreads along the chromosome to create chromosome loops.

---

## [Decision Letter]

**Decision letter after peer review:**

Thank you for submitting your article "Condensin DC spreads linearly and bidirectionally from recruitment sites to create loop-anchored TADs in *C. elegans*" for consideration by *eLife*. Your article has been reviewed by 3 peer reviewers, and the evaluation has been overseen by a Reviewing Editor and Jessica Tyler as the Senior Editor. The reviewers have opted to remain anonymous.

The reviewers and the Reviewing Editor agree that the manuscript has potential, but extensive revision and new experiments and analysis are needed to make the case for publication in *eLife* and move forward. The reviewers raised concerns about (i) the novelty of the study (as compared to Albritton et al. 2018, and Anderson et al. 2019, and other studies); and (I) validity of the conclusions drawn from presented data. Nevertheless, the study is unique in its approach of inserting rex sites into largely DCC-free autosomes, and can be improved and developed to become an impactful publication.

Reviewers and the Reviewing Editor feel that suggested mechanisms, i.e. (i) loading of DCC at rex sites, and (ii) function of rex sites as extrusion boundaries, cannot fully explain presented data. While ChIP-seq data support the loading of DCC but less of its spreading, the Hi-C data indicates the boundary role and shows no indication of the loading. I.e. Hi-C and ChIP-seq data are hard to reconcile with each other. If DCC indeed loads at integrated rex sites, one would expect to see accumulation at these sites in ChIP-seq (that may to some extent be present) and stripes emanating from rex sites on Hi-C as observed at condensin loading sites in bacteria (in the case of bidirectional extrusion) or as computed in simulation (Banigan et al. *eLife* 2020;9:e53558). Such stripes are clearly absent in presented data, putting the loading at rex in question. Reviewers didn't find the author's explanations coherent and convincing.

Essential Revisions:

1) If the authors want to pursue the double-function of rex sites they need to find support for the loading process in Hi-C. Potential ways of doing this would be to generate Hi-C data for single insertions. To increase the effect, authors may want to insert a strong rex site into an autosome. Or insert many rex sites (~5-10) spaced far from each other and examine an average Hi-C map at rex sites. Deeper sequencing and high-quality Hi-C (new MNase-involving or similar protocols) can help to reveal patterns of loading. Horizontal or 45-degree lines or other structures emanating from rex sites would be indicative of loading.

(*) The Reviewing Editor would also recommend considering a possibility of depleting non-DCC condensin I (or other suspected SMCs) that may be abundant at autosomes and masks the effect of DCC loading. In fact, abundance of another SMC may explain apparent disagreement between ChIP-seq and Hi-C: i.e. rex sites of DCC loading would block extrusion by another SMC, while having little effect on Hi-C maps otherwise. It's presence could be quantified by IP for another component of the condensin complex. And relative abundance of DCC and non-DCC condensin I could reveal loading.

2) The authors should consider the alternative that peaks on DCC ChIP-seq are mere reflection of stopping at rex rather than loading there. Demonstrating spreading of DCC from rex is essential. The Reviewing Editor suggests examining the level of DCC at non-rex sites, proximal to rex: an increasing level of DCC at such non-rex sites upon insertion of more and more potent rex sites would support loading at rex. Perhaps a series of single-site insertions could be better than inserting two sites as they could be mutually blocking each other.

3) Analysis: ChIP-seq analysis and quantification appears to have several issues. The scale on ChIP-seq plots appear to be insistent and disagree with each other (Figure 1A: Y scale goes to 1.5 and peaks at rex are <1.5, while in the inset up to 3, and several peaks exceed 3.) Figure S1A is more telling as it shows X-chromosome on the same scale. Quantitative ChIP-seq (with spike-in) may be the best solution for comparative analysis, but authors could use signal (peaks or total) on X-chromosome for normalization of non-X signals, making non-X ChIP-seq data comparable with each other. Figure 3B ChIPseq also appears all on different Y-scales, normalizing the distal part of X-chromosome can help with comparative analysis. Similarly, reviewers were concerned about quantification of 4C, its agreement with Hi-C. Analysis on Figure 4B wasn't clear and raised several concerns. It was also surprising that visual inspection of Hi-C in Figure 4A doesn't indicate or support findings in Figure 4B.

4) The reviewers raised concerns with results and interpretation of dCas9-targeted boundary. The lack of the insulation on Hi-C argues against not in favor of the stopping function of dCas9-recruited proteins. Accumulation of DCC without insulation would argue for either (a) loading of DCC at the DNA that because more accessible due to nucleosome eviction by the dCas9-recruitment, and/or (b) over-ChIPability of such regions when they become more accessible -- a well-known artifact of ChIP-seq.

5) Presentation: presenting Hi-C in grayscale is suboptimal as colors would allow one to see the dynamic range of the data. In addition, statistics for Hi-C need to be provided.

*Reviewer #1 (Recommendations for the authors):*

It would be interesting to see more quantification of the effects of condensin recruitment and spreading on the genome conformation. Particularly, Figure 2 may benefit from (a) a ratio map of contact frequencies in 3 rex/WT and (b) a plot of average contact frequency vs genomic separation in the affected region. In Figure 4, a zoom-in on the affected portion of the chromosome V (17.5-20Mb of the fusion chromsome) may more convincingly support the claims. Finally, in my experience, the curve in Figure 4B seems surprisingly noisy for the quality and amount of generated data. I would recommend the authors to double check the procedure by which they generated these curves.

This manuscript may benefit from a stronger and more focused statement on how it improves the state of the knowledge in the field. Many of the claims – insulation by individual rex sites, formation of loops/dots between rex sites on Hi-C maps, spreading of condensins on autosomes fused with the chromosome X – were already made in or could be inferred from the previous studies (e.g. Anderson et al. 2019, Crane et al. 2015). Of course, the authors do cite these studies in a complete and honest fashion; moreover, if I understand correctly, this study for the first time showed the effect of rex site insertions into a "clean slate" of an autosome, which is an important achievement. However, for a person who is not deeply engaged with the field of dosage compensation in *C. elegans*, the novelty of this paper may not be immediately obvious.

---

## [Author Response]

The reviewers and the Reviewing Editor agree that the manuscript has potential, but extensive revision and new experiments and analysis are needed to make the case for publication in eLife and move forward. The reviewers raised concerns about (i) the novelty of the study (as compared to Albritton et al. 2018, and Anderson et al. 2019, and other studies); and (I) validity of the conclusions drawn from presented data. Nevertheless, the study is unique in its approach of inserting rex sites into largely DCC-free autosomes, and can be improved and developed to become an impactful publication.

The novelty of the study:

Our work combines the strengths and addresses the weaknesses of models that were put forward in previous publications and tests several predictions using ectopic insertion experiments, presenting a new model that reconciles contradictory observations from previous work. More specifically:

Compared to qChIP data in Albritton et al. 2018 (Figure 6), our work adds a key experimental test for loop extrusion hypothesis (the *super rex* insertion on chromosome-II) and provides ChIP-seq data in the insertions, which (unlike qChIP) enabled analysis of condensin DC spreading (see section on Condensin DC is loaded at rex sites and spreads in either direction). Albritton et al. 2018 demonstrates the ectopically inserted rex sites cooperatively enhances binding at *rex*. Here we demonstrate that their cooperative function is also important for spreading and changes in 3D structure of DNA.

Compared to the *rex* deletion or *rex* insertion on X-chromosome in Anderson et al. 2019, our work shows the sufficiency of *rex* sites, when functioning as a cooperative unit, to recapitulate features of the X-chromosome TADs on autosome (see section on Ectopic insertion of rex elements leads to condensin DC spreading and formation of loop-anchored TADs). While the similar attempt was made in Anderson et al. 2019 (Figure S4), the authors inserted *rex* sites megabase apart from each other on chromosome-I, resulting in minimal binding and no changes in Hi-C matrix, likely due to the lack of cooperation between *rex* sites.

Validity of the conclusions drawn from presented data:

The reviewers were concerned by the lack of a stripe/flame emanating from the *rex* sites. The original Hi-C data using mixed developmental stage embryos (also used in Anderson et al. paper) provides patterns from a more heterogeneous cell population compared to staged L3 larvae, which we added to this manuscript where Hi-C patterns are ‘sharper.’ This led to the observation of ‘stripes’ and TADs (Figure 1). In the text we discuss the results and interpretation of each of our ectopic insertion experiments and published data in light of different models of loop extrusion to clarify the validity of the conclusions drawn from presented data. At the end, we add a model that condensin DC is a one-sided LEF with a movable inactive anchor (see section on A model to explain X-specific recruitment of condensin DC and formation of loop-anchored TADs by rex sites). Aside from the limitation of Hi-C resolution, our reasoning is in agreement with the simulation in Banigan et al. 2020.

Reviewers and the Reviewing Editor feel that suggested mechanisms, i.e. (i) loading of DCC at rex sites, and (ii) function of rex sites as extrusion boundaries, cannot fully explain presented data. While ChIP-seq data support the loading of DCC but less of its spreading, the Hi-C data indicates the boundary role and shows no indication of the loading. I.e. Hi-C and ChIP-seq data are hard to reconcile with each other. If DCC indeed loads at integrated rex sites, one would expect to see accumulation at these sites in ChIP-seq (that may to some extent be present) and stripes emanating from rex sites on Hi-C as observed at condensin loading sites in bacteria (in the case of bidirectional extrusion) or as computed in simulation (Banigan et al. eLife 2020;9:e53558). Such stripes are clearly absent in presented data, putting the loading at rex in question. Reviewers didn't find the author's explanations coherent and convincing.

To summarize, the reviewers comment that the model of *rex* sites acting as loading (i) and barrier (ii) sites cannot fully explain data. We argue that:

1) ChIP-seq data from *rex* insertions does not result in spreading: Because robust recruitment of condensin DC requires multiple strong *rex* sites, we engineered a more potent version of an existing *rex* site by stringing together six truncated *rex* elements, which we call *super rex*. In this strain, spreading from a point source is clearly apparent and diminishes with increasing distance away from the insertion site. This spreading is also observable in conditions with more than one *rex* inserted (Figure 5, Supplemental Figure 5). We surmise that the lack of spreading in the original data had more to do with the initial ChIP analysis, mapping reads to the normal reference genome (here we made “corresponding edited genomes with the inserted sequence included” and not using a normalization method that minimized technical variation between replicates (here we used scaling to make ChIP data between strains comparable to each other). Please see the methods section for updated analyses.

2) Hi-C lack of stripe: In three *rex* insertion and *super rex* insertion, the stripe is weak but visible, particularly when data is analyzed compared to the wild type control (Figure 6). In addition, the stripes from averaged Hi-C data plotted across strong *rex* sites are more visible in the L3 at the endogenous X chromosome (Figure 1). The lack of stripes in the original data is likely due to doing Hi-C experiment in embryo where the patterns are murky, mapping to the reference genome (instead of updated genome with inserted sequence), and not providing the comparison data to wild type (division of Hi-C matrix between insertion/wildtype), and not using a more dynamic color range. All of which has been fixed in the revision (Figure 6). We thank the reviewers/editor for their specific suggestions while solving these problems.

Essential Revisions:1) If the authors want to pursue the double-function of rex sites they need to find support for the loading process in Hi-C. Potential ways of doing this would be to generate Hi-C data for single insertions. To increase the effect, authors may want to insert a strong rex site into an autosome. Or insert many rex sites (~5-10) spaced far from each other and examine an average Hi-C map at rex sites. Deeper sequencing and high-quality Hi-C (new MNase-involving or similar protocols) can help to reveal patterns of loading. Horizontal or 45-degree lines or other structures emanating from rex sites would be indicative of loading.

Generate Hi-C data for single insertions:

Previously, in Albritton et al. 2019, *rex-8* was computationally categorized as the strongest of all *rex* sites based on ChIP-seq data. We performed ChIP-seq and Hi-C in single *rex* (*rex-8*) insertion on chromosome-II, where there was no spreading and no changes in Hi-C matrix. This is in agreement with Anderson et al. 2019 (Figure S4), where the authors inserted 3 *rex* sites spaced a megabase apart from each other only to observe no changes on Hi-C matrix. Thus, the problem with inserting a single strong *rex* is that it is not enough to create a domain of condensin DC binding on an autosome. To address this problem, we devised a stronger *rex* (see also response to summarized editor comments above).

In revision, we generated a *super rex* containing six 150bp truncated strong *rex* sites, which gave a robust spreading and observable stripes emanating to both sides from the insertion site. While we note that *super rex* is not a naturally occurring DNA element, it is simply a combination of existing *rex* sites, which do not have a clear definition other than they harbor one or more of the 12-bp motifs. We attribute the lack of ‘spreading’ or effect on Hi-C data in single *rex* insertion experiments to the autosomes being outcompeted by the X-chromosome for access to condensin DC (see in Discussion).

(*) The Reviewing Editor would also recommend considering a possibility of depleting non-DCC condensin I (or other suspected SMCs) that may be abundant at autosomes and masks the effect of DCC loading. In fact, abundance of another SMC may explain apparent disagreement between ChIP-seq and Hi-C: i.e. rex sites of DCC loading would block extrusion by another SMC, while having little effect on Hi-C maps otherwise. It's presence could be quantified by IP for another component of the condensin complex. And relative abundance of DCC and non-DCC condensin I could reveal loading.

Comment on a non-DCC SMC complex masking the effect of loading/depleting non-DCC SMC complexes:

The potential interactions between different SMC complexes is a new avenue of research that we are currently pursuing. While we did not deplete other SMC complexes in the current study, we agree that the experiments make a powerful prediction. If condensin DC is being pushed around by another SMC complex, then removal of such protein would enhance ‘stripes’ at *rex* sites. This is incorporated into our model as a possibility (Figure 7), but we would like to keep this as part of the future direction of our study.

Comment on *rex* sites of DCC loading would block extrusion by another SMC:

*Rex-8* insertion on chromosome-II is an interesting situation where there is condensin DC binding at the *rex* but no spreading. Here, binding without spreading resulted in no observable insulation effect on Hi-C, suggesting that condensin DC binding at *rex* sites do not function as indiscriminate barriers to other SMC complexes on autosomes.

Comment on relative abundance of DCC and non-DCC condensin I could reveal loading:

Given the conclusion that *rex-8* does not act as a boundary, we did not ChIP for other SMC complexes. Note that DPY-27 is a subunit unique to condensin DC and is not part of condensin-I/II (Csankovszki 2009).

2) The authors should consider the alternative that peaks on DCC ChIP-seq are mere reflection of stopping at rex rather than loading there. Demonstrating spreading of DCC from rex is essential. The Reviewing Editor suggests examining the level of DCC at non-rex sites, proximal to rex: an increasing level of DCC at such non-rex sites upon insertion of more and more potent rex sites would support loading at rex. Perhaps a series of single-site insertions could be better than inserting two sites as they could be mutually blocking each other.

Demonstrating spreading from *rex* by analyzing insertion of more and more potent single *rex* site:

We performed ChIP-seq upon insertion of weaker (*rex-1*), strong rex (*rex-8*) and *super rex* (an array of 6 strong *rex*), which showed no recruitment, recruitment and recruitment and spreading respectively (see section on Condensin DC is loaded at rex sites and spreads in either direction). The reviewer’s prediction that addition of more potent *rex* resulting in increasing DCC level at surrounding regions is qualitatively met. A conceptually similar prediction that adding more *rex* would increase DCC level at surrounding regions is also met t (Figure 5 and Supplemental Figure 5-1).

3) Analysis: ChIP-seq analysis and quantification appears to have several issues. The scale on ChIP-seq plots appear to be insistent and disagree with each other (Figure 1A: Y scale goes to 1.5 and peaks at rex are <1.5, while in the inset up to 3, and several peaks exceed 3.) Figure S1A is more telling as it shows X-chromosome on the same scale. Quantitative ChIP-seq (with spike-in) may be the best solution for comparative analysis, but authors could use signal (peaks or total) on X-chromosome for normalization of non-X signals, making non-X ChIP-seq data comparable with each other. Figure 3B ChIPseq also appears all on different Y-scales, normalizing the distal part of X-chromosome can help with comparative analysis. Similarly, reviewers were concerned about quantification of 4C, its agreement with Hi-C. Analysis on Figure 4B wasn't clear and raised several concerns. It was also surprising that visual inspection of Hi-C in Figure 4A doesn't indicate or support findings in Figure 4B.

Analysis ChIP-seq scales inconsistent:

The original ChIP scores did not do further normalization to deal with the variability between the biological replicates of ChIPs, particularly in embryos. To address this problem, we used the mean signal on autosomes (excluding chromosome-II where *rex* sites are inserted) and the X-chromosome to rescale the input normalized ChIP-seq data such that the mean of autosomes is 0 and the mean of the X is 1 (see method section mRNA-seq and ChIP-seq). Similar method was used for X-V fusion. This normalization recapitulates the patterns in non normalized data while taking care of the values on the y axis such that they are similar between biological replicates. The revised manuscript provides the details of the analysis and better figures representing the ChIP data.

Hi-C analysis concerns regarding Hi-C quantification and visualization of XV fusion:

We did additional analyses of the X;V data. In revision, we provide log2ratio plot, which shows increased ‘shorter-range’ interactions in the XV fusion strain that is the most pronounced on the chromosome-V side of the fusion where the spreading occurs. We also adopted the method of computing the log-derivative of log-binned P(s) to infer mean loop size (see section on Spreading of condensin DC entails loop extrusion but cannot sufficiently form TADs without rex sites). The quantification shows that the mean loop size of the fusion side becomes similar to that of the mean loop size of the X-chromosome (Figure 2C). We have also included compartment analysis, which shows decreased compartmentalization specifically at the autosomal spreading region (Figure 2D), and suggested increased loop extrusion at this area due to condensin DC spreading.

4) The reviewers raised concerns with results and interpretation of dCas9-targeted boundary. The lack of the insulation on Hi-C argues against not in favor of the stopping function of dCas9-recruited proteins. Accumulation of DCC without insulation would argue for either (a) loading of DCC at the DNA that because more accessible due to nucleosome eviction by the dCas9-recruitment, and/or (b) over-ChIPability of such regions when they become more accessible -- a well-known artifact of ChIP-seq.

The lack of the insulation on Hi-C in dCas9-targeted loci:

We ran into some technical difficulties working with the strain expressing sgRNA for dcas9-targeting. The worms are sick and hard to grow. This resulted in sub-optimal Hi-C and ChIP-seq data. However, computing log2ratio of the matrix and insulation scores, we note a slight increase in the insulation effect at the targeted locus relative to the surrounding.

Loading of DCC/over-ChIPability due to accessibility:

We performed IgG ChIP, and as the reviewer had predicted, the targeting of dCas9 resulted in the loci becoming hyper-chippable. We also show that this is not an artifact of cross-reactivity with protein A or G. This ChIP-seq artifact could not be computationally ‘removed’ by division, as IgG signal at the loci is much greater than that of DPY-27 (see section on A dCas9-based block failed to recapitulate rex-like boundary on the X-chromosome). We thus refrained from making any conclusions about condensin DC binding at the block.

5) Presentation: presenting Hi-C in grayscale is suboptimal as colors would allow one to see the dynamic range of the data. In addition, statistics for Hi-C need to be provided.

We followed the suggestion and used the ‘fall’ colormap for visualization of Hi-C data. The statistics for individual replicates are provided in the Supplementary file 1 (please check different tabs on the excel file).

Reviewer #1 (Recommendations for the authors):It would be interesting to see more quantification of the effects of condensin recruitment and spreading on the genome conformation. Particularly, Figure 2 may benefit from (a) a ratio map of contact frequencies in 3 rex/WT and (b) a plot of average contact frequency vs genomic separation in the affected region. In Figure 4, a zoom-in on the affected portion of the chromosome V (17.5-20Mb of the fusion chromsome) may more convincingly support the claims. Finally, in my experience, the curve in Figure 4B seems surprisingly noisy for the quality and amount of generated data. I would recommend the authors to double check the procedure by which they generated these curves.

Figure *rex* insertion (Figure 6 in revised manuscript):

a) Ratio map insertion/wt

The ratio plots are provided for all insertions

(b) Plot average contact frequency vs genomic separation

We have come to learn that P(s) curves in such a small submatrix could be noisy. However, we do provide insulation scores and their subtraction against wildtype in each Hi-C figure.

Figure XV fusion (Figure 2 in revised manuscript):

(a) Zoom in on the affected portion:

We provide an inset for region close to the fusion site, which shows changes in compartmentalization. We also provide log2ratio of fusion compared to wild type.

b) The curve seems noisy

We’ve adopted log-derivative plots and use of larger sub-matrix (also see response to Essential Revisions (2) on Hi-C analysis)

This manuscript may benefit from a stronger and more focused statement on how it improves the state of the knowledge in the field. Many of the claims – insulation by individual rex sites, formation of loops/dots between rex sites on Hi-C maps, spreading of condensins on autosomes fused with the chromosome X – were already made in or could be inferred from the previous studies (e.g. Anderson et al. 2019, Crane et al. 2015). Of course, the authors do cite these studies in a complete and honest fashion; moreover, if I understand correctly, this study for the first time showed the effect of rex site insertions into a "clean slate" of an autosome, which is an important achievement. However, for a person who is not deeply engaged with the field of dosage compensation in *C. elegans*, the novelty of this paper may not be immediately obvious.

Please see our comments to the editor above addressing novelty and our new Discussion section summarizing the contradictions and shortcomings of previous models.